# Biomechanical influences of gait patterns on knee joint: Kinematic & EMG analysis

Jin Ju Kim[1]ᵒ, Han Cho[1]ᵒ, Yulhyun Park[2], Joonyoung Jang[2], Jung Woo Kim[3], Ju Seok Ryu[2,4]*

**1** Department of Medicine, Seoul National University College of Medicine, Seongnam-si, Republic of Korea, **2** Department of Rehabilitation Medicine, Seoul National University Bundang Hospital, Seongnam-si, Republic of Korea, **3** Department of Rehabilitation Medicine, Seoul National University Hospital, Seongnam-si, Republic of Korea, **4** Department of Rehabilitation Medicine, Seoul National University College of Medicine, Seongnam-si, Republic of Korea

ᵒ These authors contributed equally to this work.
* jseok337@snu.ac.kr

**Data Availability Statement:** All relevant data are within the paper and supporting information files.

**Funding:** This research was supported by a grant of the Korea Health Technology R&D Project through the Korea Health Industry Development

## Abstract

### Background

As lumbar spinal stenosis commonly occurs between the L2 and L5 segments, hip abductors are easily affected. However, studies regarding the gait pattern in these patients from the coronal plane have not yet been conducted.

### Purpose

To determine the effects of lumbar spinal stenosis on the gait pattern (stride width and femorotibial angle) and hip abductor surface electromyography in varied stride widths compared with healthy individuals.

### Study design

Prospective case-control study.

### Methods

Seventeen patients and 20 healthy individuals were enrolled. Each participant completed three gait assessments in their normal gait, adducted gait and abducted gait. The femorotibial angle and surface electromyography signals were measured. Pain scores was used to quantify the degree of discomfort in the gluteal area and medial side of the knee.

### Results

When the hip abductors' surface electromyography signals were normalized by quadriceps femoris, patients group showed significantly higher activation ratios throughout all gait patterns. Generally, surface electromyography signals and ratios were significantly higher during abducted gait compared with a normal gait. Femorotibial angle became significantly closer to the varus in healthy individuals during abducted gait. When femorotibial angle during normal gait was compared between the two groups, patients group exhibited slightly

Institute (KHIDI), funded by the Ministry of Health & Welfare, Republic of Korea (grant number: HI18C1169).

**Competing interests:** The authors have declared that no competing interests exist.

wider stride width and FTA significantly closer to the varus. Pain scores were significantly higher in the patient group and during abducted gait.

## Conclusion

Wider stride widths indicated increased relative activation of the hip abductors, closer proximity between femorotibial angle and varus, and increased pain scores for discomfort. The same tendency was observed in patients group when compared with healthy individuals. Widening of stride width in patients group despite abductor weakness suggests that additional muscle recruitment may be needed to maintain balance. Furthermore, such a distinctive gait pattern exerts increased loading on the medial knee, relating to the escalated risk of degenerative knee osteoarthritis.

## Introduction

Lumbar spinal stenosis (LSS) is defined as a diminished space in the lumbar spinal canal, resulting in symptoms caused by compressed neural and vascular elements in the lumbar spine [1, 2]. Due to compressed neural and vascular elements, LSS may cause pain in the gluteal area and the lower extremities, as well as fatigue and back pain, among others [2–4]. LSS is clinically diagnosed based on radiologic and clinical criteria; however, since MRI is considered as the "most appropriate, noninvasive test to confirm the presence of anatomic narrowing of the spinal canal or the presence of nerve root impingement," radiologic findings are accepted as the most appropriate diagnostic method.[2]

The narrowing of the spinal canal is typically caused by degenerative changes [5]; therefore, LSS commonly affects the elderly, and its incidence increases in aging populations [6, 7]. Walking is a complex task [8], and the ability to walk declines with age [9]. Since LSS symptoms are aggravated with exercise, a decrease in walking ability, as aforementioned, is accelerated with LSS [4, 6], significantly affecting the daily life of LSS patients. Moreover, LSS is one of the most significant causes of gait restriction in elderly patients, especially for those over the age of 55 years, along with other clinical conditions, including osteoarthritis [10].

Stride length is also commonly measured in studies regarding gait patterns of LSS patients [11, 12]. Such studies approach gait patterns of LSS patients from the sagittal plane. However, to the best of our knowledge, studies regarding the gait pattern of LSS patients from the coronal plane have not yet been conducted.

In normal gait in coronal plane torque, adducting vector is maintained and the intensity of gluteus medius, minimus and tensor fascia lata muscles increases to 20%, 20% and 25% of maximal manual muscle test value, respectively.[13] As LSS commonly occurs between the L2 and L5 segments, hip abductors, innervated at L4 and L5, are easily affected; therefore, weakness of hip abductors develops in severe cases and affects walking in the coronal plane. [14]

We hypothesize that LSS patients can develop hip abductor weakness and sensory change, and these changes may induce abnormal gait patterns in the coronal plane. Moreover, abnormal gait pattern in the coronal plane could affect the mechanics of the knee joints and the development of knee osteoarthritis.

The purpose of this study was to identify the changes in kinematic and electromyographic aspects of gait patterns in LSS patients compared with healthy individuals, thereby verifying the causative effects of LSS on the development of knee osteoarthritis in biomechanical aspect.

## Materials and methods

### Study design

This prospective case-control study was performed between September 2018 and February 2019 in the tertiary hospital. All patients provided written informed consent, and the study protocol was approved by the Institutional Review Board (IRB No.: B- 1810-497-309).

### Participants

The inclusion criteria for LSS were as follows: independent ambulation, age greater than 50 years, grade greater than 1 with the Hufschmidt grade (symptomatic criteria),[15] and more than moderate stenosis at L2-5 segments with MRI criteria.[16] A total of 17 patients were enrolled in the LSS group, and 20 healthy adults free of LSS symptoms (numbness and tingling of the lower extremities) were recruited to be included in the control group.

### Stride width

Stride width was defined as the medial-lateral distance between the heels. Stride width was measured using a baropodometric platform. Data were collected using the *FreeStep* software[R] (Fig 1(A), Sensor Medica, Rome, Italy). This software automatically demonstrated the step length, stride length and foot progression angle which is the angle between the heel and the path of gait. Stride width was calculated from the data of the contact points obtained by baropodometric platform (Fig 1(B, Right)). Participants were instructed to walk back and forth as usual. This was repeated once more, resulting in a total of four trials. The first two trials were discarded as test trials, and the stride width of the last two trials was analyzed. Since optimum stride width is given as a ratio relative to leg length (0.12~0.13L; L: leg length) [17], normalized stride width values were computed by dividing stride width by height.

### Gait designation

To elucidate how stride width can affect muscle activation and limb alignment, three distinctive gait patterns were designed: normal gait, adducted gait, and abducted gait (Fig 1(B, Right)). During normal gait, participants were asked to walk as usual. Then, they were asked to walk with medial borders of their feet touching each other, which was defined as the adducted gait. Finally, they were requested to walk with medial borders of their feet approximately 40cm apart, which was termed as the abducted gait. For each gait pattern, individuals were asked to walk for approximately 6 meters and repeat the gait twice, resulting in a total of six gait trials and the mean values were calculated.

### Surface electromyography analysis

The sEMG signals were measured in the left and right gluteus medius (GMe), tensor fasciae latae (TFL), and quadriceps femoris (QF), using the wireless sEMG analysis system (BTS FREEEMG 1000 with EMG-BTS EMG-Analyzer[R] (BTS Bioengineering Co, Italy, Fig 1(C)). Participants were asked to walk in the three aforementioned instructed gait patterns with electrodes placed, following the guidelines of *Cram's Introduction to Surface EMG* [18]. To measure the activation of GMe, electrodes were placed parallel to the muscle fibers, on the proximal third of the line, between the iliac crest and greater trochanter. The activation of TFL was recorded by electrodes placed parallel to the muscle fiber, 2cm below the anterior superior iliac spine. Electrodes for measuring the activation of QF were placed parallel to the muscle fibers, halfway between the knee and the iliac spine.

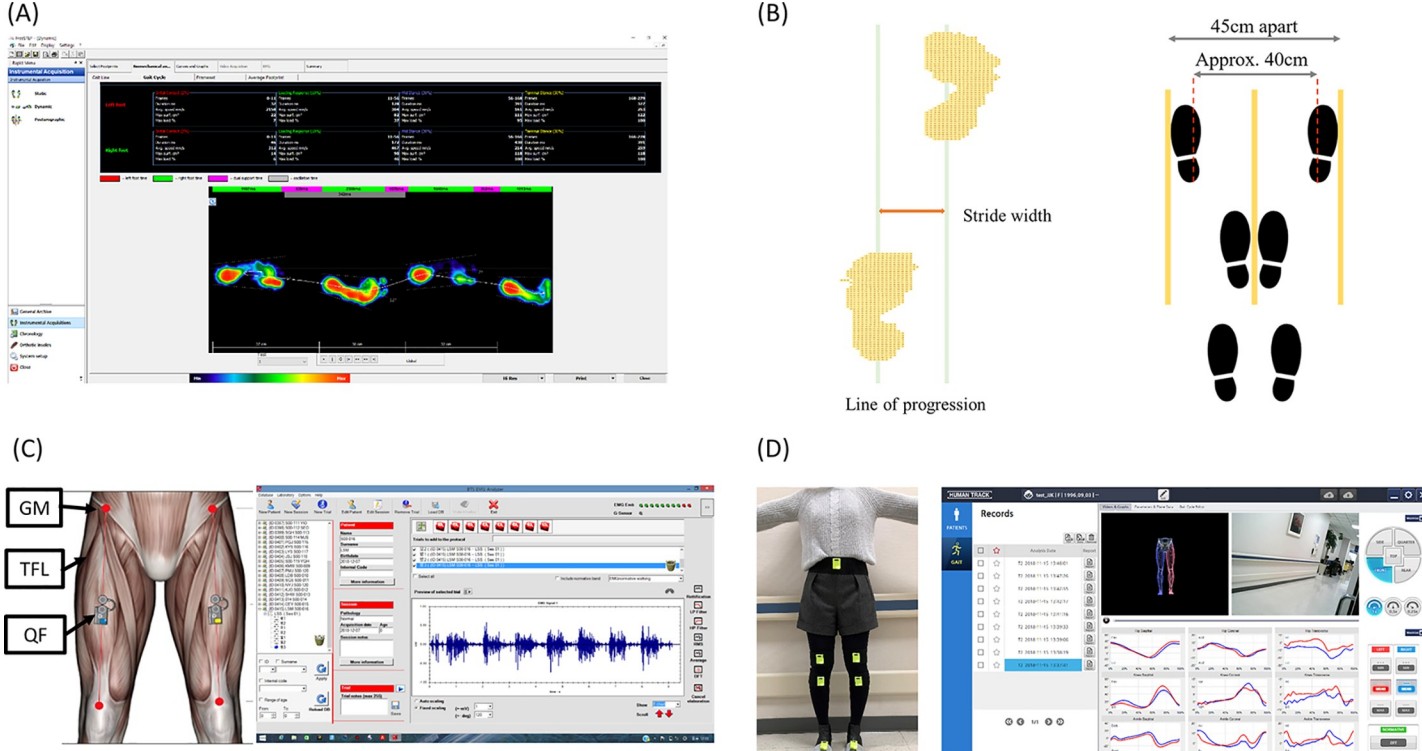

**Fig 1. These figures show the evaluation method.** (A) Stride width was measured using the pressure plate with a sensorized matrix (*FreeStep*). Participants were instructed to walk end-to-end for 4 times. The first 2 trials were regarded as familiarization trials and the outcomes from the last 2 trials were obtained for step width analysis. (B) Stride width was defined as the distance between your heels when each heel is at its lowest point during the stride. The corresponding coordinates were collected and analyzed through manual specified by FreeStep software. Raw data was exported as Microsoft Excel file (.XLS format). The Microsoft Excel file containing each contact point (in yellow). The progression line is in green and the stride width is marked with an orange arrow. The width of each cell is 0.5cm, thus the number of cells between the green lines was counted and the final stride width was calculated (Left image). Three gait patterns (Right image). First, the participants were instructed to walk in their normal pace ("Normal gait"). Then, they were asked to walk with their medial borders of feet touching each other ("Adduction gait"). Finally, they were told to walk with their feet approximately 40cm apart ("Abduction gait"). (C) Surface electromyography (sEMG) analysis. sEMG was measured using BTS FREEEMG 1000 with EMG-BTS EMG-Analyzer®, with electrodes placed based *on Cram's introduction to surface EMG*–proximal 1/3 between the iliac crest and the greater trochanter for GM, approximately 2cm below the ASIS for TFL and approximately 1/2 between the knee and the iliac spine for QF. sEMG signals of the muscles were measured twice for each gait pattern. (D) Gait analysis in the coronal plane. The femorotibial angle (FTA) was measured using Human Track® (Gait & Motion Analysis System). The lumbar sensor was placed on the midpoint of the lateral iliac crest. The thigh and the shank sensors were placed 9cm proximal and distal from the patella respectively. The foot sensor was placed on the 2nd and the 3rd metatarsal. FTA was calculated from the midstance phase peaks in the knee coronal plane.

Using the recorded sEMG signals, the root-mean-square (RMS) value and peak value were measured. The signals recorded during initiation and termination of each gait trial were excluded. Also, the peak activation and RMS values during gait were obtained and compared among the three gait patterns. These measurements were performed two times, and the mean values were calculated. To decrease the individual variation of sEMG, the sEMG ratios (sEMG values divided by the QF muscle at each cycle) were compared among the gait patterns.

## Gait analysis in the coronal plane

To assess limb alignments, Femorotibial angle (FTA) was measured using the Human Track®, Gait & Motion Analysis System (RBIOTECH CO., LTD, Korea, Fig 1(D)). Straps were attached according to the anatomical landmarks; the lumbar sensor was placed on the midpoint of the lateral iliac crest; the thigh and shank sensors were placed 9cm proximal and distal from the patella, respectively; and the foot sensors were placed on the 2nd and the 3rd metatarsal bones. FTA was obtained from the analysis of the coronal plane knee joint. Gait cycles of

each trial from the initiation phase to the termination phase were summed up. The abnormal gait cycles were also discarded. After summation, the peak FTA during the mid-stance phase was obtained; the mid-stance phase was defined as 10–40% of the raw gait cycle. These measurements were performed two times, and the mean values were calculated.

### Subjective perception of discomfort during gait

Between the repeated trials of each gait, participants were asked to rate their subjective perception of discomfort using the visual analog scale (VAS). The discomforts of the hip and medial side of the knee were separately reported during each of the three designated gait patterns. Participants expressed their discomfort on a scale from 0 to 10, where 0 indicated no discomfort and 10 indicated most severe discomfort.

### Statistical analysis

All statistical analyses were conducted using SPSS version 25 (SPSS Inc.; Chicago, IL, USA). Between the control and LSS groups, an independent T-test was conducted to compare the means (when P-value from Kolomogorov-Smirnov's test of normality was < 0.05, independent samples Mann Whitney U test was conducted). When measurements from normal gait were compared with those of abducted and adducted gait within the group, repeated-measures ANOVA was used to compare the means. Bonferroni correction was used to reduce the risk of type I error. P-value < 0.05 was considered to be statistically significant.

## Results

Demographic data are presented in Table 1. The average age was 66.1 ± 8.0 years in the LSS group and 50.5 ± 6.0 years in the control group. The gender ratios were not significantly different between the two groups (p-value > 0.05). The average height and weight were 157.6 ± 6.3 (cm) / 59.3 ± 8.4(kg) and 163.1 ± 6.4(cm) / 60.9 ± 11.8(kg) in the LSS group and the control group, respectively. The average normalized stride width was slightly greater in LSS patients (4.11%) compared with that in the control group (4.07%) but without statistical significance.

Table 2 illustrates the findings of gait analysis in the coronal plane. Lt. side FTA significantly changed to the varus when the control group walked in an abducted gait (p-value < 0.05). The normal gait in the LSS group showed a significant varus angle than those in the control group (p-value < 0.05). Other parameters between the two groups or gait patterns were not significantly different (p-value > 0.05, Fig 2).

Table 3 and Fig 3 illustrate the RMS and peak sEMG values from GMe, TFL, and QF. RMS and peak values of GMe, TFL, and QF were significantly different in three gait patterns in both groups. In the control group, the RMS and peak values of TFL were significantly increased during abducted gait than normal and adducted gaits (p-value < 0.05). The RMS and peak values of GMe were significantly increased during abducted gait than adducted gait (p-value < 0.05) and showed increased tendency than normal gait (p-value, RMS: 0.061, peak: 0.074). In the LSS group, RMS and peak values of GMe were significantly increased during abducted gait than adducted and normal gaits (p-value < 0.05). The RMS and peak values of TFL showed an increased tendency during abducted gait than normal gait (p-value, RMS: 0.052, peak: 0.061). Also, the RMS and peak values of GMe and TFL were significantly increased during adducted gait than normal gait (p-value < 0.05). When we compared QF among the three gait patterns, the RMS values of QF during abducted gait were significantly decreased in the control group (p-value < 0.05), while a significant increase was observed in the LSS group (p-value < 0.05). In comparison between the two groups, the RMS and peak values of QF were significantly lower during normal and adducted gaits, and the RMS and peak

**Table 1. Demographic data of the lumbar spinal stenosis group and the control group.**

| | LSS group | | Control group | | |
| --- | --- | --- | --- | --- | --- |
| | Sample size | Mean | Sample size | Mean | *p*-value |
| Age (years) | 17 | 66.1±8.0 | 20 | 50.5±6.0 | 0.000 |
| Height (cm) | 17 | 157.6±6.3 | 20 | 163.1±6.4 | 0.007 |
| Weight (kg) | 17 | 59.3±8.4 | 20 | 60.9±11.8 | 0.647 |
| Gender (M/F) | 3/14 | - | 3/17 | - | 0.774 |
| Normalized stride width (%) | 17 | 4.113±1.995 | 20 | 4.067±1.927 | 0.537 |

LSS group: Lumbar spinal stenosis group. Values are expressed as mean ± standard deviation.

values of TFL were significantly higher during normal, adducted and abducted gaits in the LSS group than those of the control group (p-value < 0.05).

Table 4 illustrates the sEMG RMS ratios (sEMG values divided by the QF muscle at each cycle). The RMS ratio of GMe and TFL in the control group and the RMS ratio of GMe in the LSS group showed a significant increase during the abducted gait than the normal gait (p-value < 0.05). (p-value < 0.05) When compared between the two groups, all sEMG RMS ratios were higher in LSS patients compared with the control group, regardless of stride width (p-value < 0.05). The sEMG peak ratio showed similar findings to the results of the sEMG RMS ratio (Table 4).

The discomfort levels according to the gait pattern are described in Table 5 and Fig 4. Compared with the normal gait, the VAS scores were significantly higher during the adducted and abducted gait, which were also significantly higher in both the hip and medial sides of the knee (p-value < 0.05), except for the VAS score of hip in the LSS group, which showed an insignificant increase (p-value > 0.05). The overall VAS scores in the LSS group were higher than those in the control group (p-value < 0.05).

## Discussion

Most of the previous studies to examine gait patterns in LSS patients focused on decreased physical performance[4, 6]. Among few exceptions, changes in stride or joint angles have only been viewed from the sagittal plane[19–23]. Though hip abductors play a significant role during the single-limb support phase of the gait cycle, most electromyography and gait analyses have also been focused on the sagittal plane[24–27]. Thus, specific gait patterns of LSS patients from the coronal plane are necessary to be clarified.

LSS patients are known to develop hip abductors weakness, which is mainly innervated by the lumbar 5 segments, as a result of radiculopathy.[2] With major changes in the nerves and

**Table 2. The analysis of coronal plane in gait analysis.** Femorotibial angle (FTA) changed to varus angle during abduction gait in both the lumbar spinal stenosis group and the control group.

| Gait pattern | LSS group | | | Control | | |
| --- | --- | --- | --- | --- | --- | --- |
| | Right | Left | Bilateral | Right | Left | Bilateral |
| Normal | 0.281±2.404# | 1.896±3.049# | 1.089±2.838# | -2.018±3.826 | 0.075±2.273 | -0.973±3.299 |
| Adducted | -0.158±2.540 | 1.423±2.248 | 0.633±2.506 | -1.880±4.704 | 0.708±2.702 | -0.586±4.028 |
| Abducted | 0.448±3.587# | 1.800±2.549 | 1.124±3.157# | -1.560±4.044 | 1.178±2.845* | -0.191±3.737 |

LSS group: Lumbar spinal stenosis group. Varus angle is positive and valgus angle is negative. The values are expressed as angle(˚).

#: Comparisons of the normal gait pattern between the LSS and the control groups.

*: Comparisons of gait patterns within each group. *, #: P-Value < 0.05

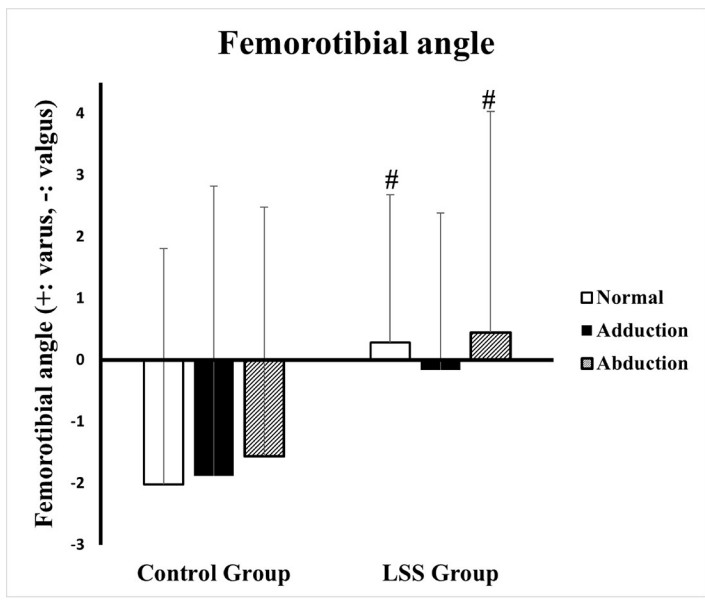
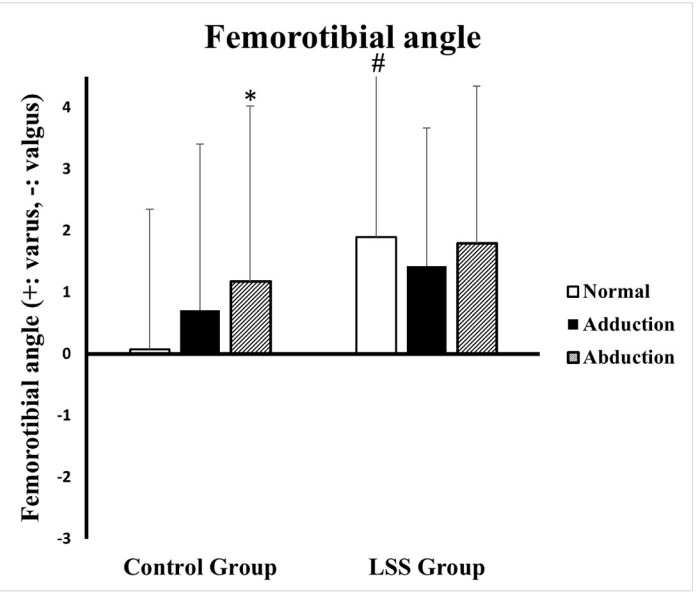

**(A) Right side**

**(B) Left side**

**Fig 2.** These figures show the results of Femorotibial angle (FTA) of the right (A) and the left side (B). The FTA was changed to varus angle during abducted gait in both groups (*, p-value < 0.05 only in the control group, left side). During normal gait, the LSS group' FTA was significantly closer to varus than the control group (#, p-value < 0.05). Varus angle is positive and valgus angle is negative. The values are expressed as angle(°).

hip abductors, alterations in gait patterns may develop especially in the coronal plane. This may lead to changes in the distribution of forces exerted on the joints affecting the bony alignment of the lower limb, predisposing patients to degenerative pathologies of the joint.[28] Thus, we hypothesized that such biomechanical changes may affect the overall gait pattern with respect to the stride width and hip abductor sEMG activation signals. Furthermore, we expected an altered gait pattern to elicit changes in the knee joint, inducing the knee into a more varus or valgus formation.

However, exact predictions were impossible with two contradictory observations. On the one hand, a wider stride width requires higher activation of the hip abductor muscles.[24] On the other, older populations with hip abductor weakness, other than LSS, were reported to

**Table 3. The surface EMG analysis of gluteus medius, tensor fascia lata and quadriceps femoris muscles.**

| Gait pattern | LSS group | | | Control group | | |
|---|---|---|---|---|---|---|
| **RMS** | GMe | TFL | QF | GMe | TFL | QF |
| Normal | 37.3 ± 25.5 | 70.0 ± 73.6# | 55.2 ± 34.0# | 38.6 ± 44.0 | 45.1 ± 30.2 | 78.1 ± 37.5 |
| Adducted | 39.4 ± 27.8 | 84.7 ± 104.0# | 59.0 ± 34.1# | 32.4 ± 22.0 | 49.5 ± 37.3 | 72.1 ± 37.0 |
| Abducted | 53.6 ± 36.9* | 89.2 ± 86.4# | 67.4 ± 37.1* | 46.3 ± 31.1* | 60.6 ± 40.6* | 66.4 ± 33.9* |
| **Peak** | GMe | TFL | QF | GMe | TFL | QF |
| Normal | 90.3 ± 61.5 | 154.7 ± 132.3 | 139.8 ± 91.0# | 100.3 ± 90.4 | 124.4 ± 79.7 | 209.4 ± 92.9 |
| Adducted | 99.4 ± 68.1 | 178.8 ± 160.3# | 144.3 ± 83.0# | 90.8 ± 57.9 | 129.6 ± 98.5 | 196.4 ± 93.7 |
| Abducted | 116.4 ± 71.0* | 175.8 ± 129.2 | 149.7 ± 83.3* | 120.7 ± 79.0* | 145.7 ± 97.5* | 170.8 ± 80.2* |

LSS group: Lumbar spinal stenosis group; RMS: Root mean square; GMe: Gluteus medius; TFL: Tensor fascia lata

*: P-Value <0.05 in comparison within each group

#: P-Value <0.05 in comparison to control group

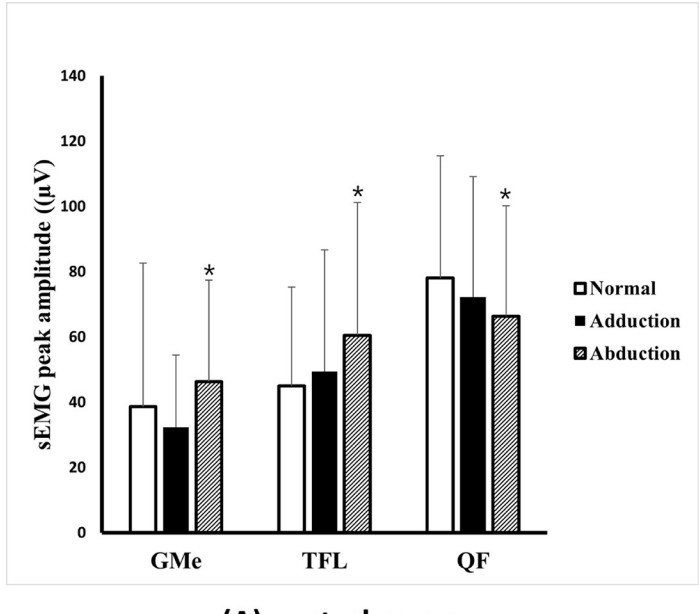
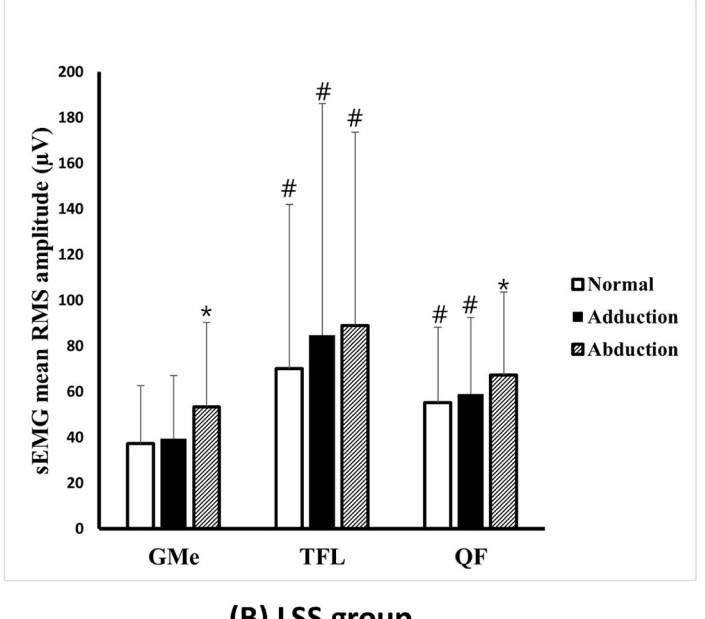

**(A) control group**     **(B) LSS group**

**Fig 3. These figures show the results of sEMG analysis of gluteus medius, tensor fasciae and latae and quadriceps femoris muscles.** (A) RMS of gluteus medius and tensor fasciae latae in the control group (A) and RMS of gluteus medius in the LSS group (B) were significantly increased during abducted gait than the normal gait. However, in the case of QF, the control group showed a decrease in amplitude (A) while the LSS group showed an increase (B, p-value < .05).

have a wider stride width.[24, 29, 30] The results of our study supported our hypothesis as LSS patients exhibited changes in their gait patterns compared with healthy individuals with respect to stride width, hip abductor sEMG signals, and FTA. LSS patients were observed to walk with a slightly wider stride width, exhibit higher hip abductor muscle activation signals, have knees closer to a varus formation, and aggravated discomfort in the hip and medial knee areas.

When instructed to walk with a wider stride width, both the control group and LSS group showed higher activation of GMe and TFL relative to QF. These consistent elevations in the sEMG signals show that the hip abductor muscles are essential during the abducted gait.

**Table 4. The RMS and peak ratio of the surface EMG (surface EMG values divided by the quadriceps femoris at each cycle) in gluteus medius and tensor fasciae latae.**

| Gait pattern | LSS group | | Control group | |
|---|---|---|---|---|
| **RMS** | GMe/nQF | TFL/nQF | GMe/nQF | TFL/nQF |
| Normal | 0.880 ± 0.814# | 1.659 ± 2.401# | 0.608 ± 0.840 | 0.634 ± 0.362 |
| Adducted | 0.836 ± 0.795# | 1.846 ± 2.878# | 0.596 ± 0.532 | 0.771 ± 0.541* |
| Abducted | 0.992 ± 0.935* | 1.738 ± 2.300# | 0.897 ± 0.845* | 1.078 ± 0.793* |
| **Peak** | GMe/nQF | TFL/nQF | GMe/nQF | TFL/nQF |
| Normal | 0.877 ± 0.852# | 1.493 ± 1.783# | 0.609 ± 0.786 | 0.672 ± 0.439 |
| Adducted | 0.885 ± 0.813# | 1.644 ± 2.024*,# | 0.643 ± 0.635 | 0.743 ± 0.524 |
| Abducted | 0.993 ± 0.880 | 1.559 ± 1.705# | 0.909 ± 0.872* | 1.042 ± 0.902* |

LSS group: Lumbar spinal stenosis group; GMe: Gluteus medius; nQF: Normalized quadriceps femoris; TFL: Tensor fascia lata: RMS: Root mean square.

*: P-Value <0.05 in comparison within each group

#: P-Value <0.05 in comparison to control group

**Table 5. The discomfort level according to gait pattern was significantly higher in the LSS group.** Discomfort increased by order of normal, adduction, abduction gait in both groups.

| Gait pattern | LSS group | | Control group | |
|---|---|---|---|---|
| | Gluteal area | Medial knee | Gluteal area | Medial knee |
| Normal | 3.294 (± 2.932)# | 2.000 (± 1.904)# | 0.000 (± 0.000) | 0.000 (± 0.000) |
| Adducted | 3.529 (± 2.875)# | 2.529 (± 2.375)*,# | 0.200 (± 0.410)* | 0.400 (± 0.598)* |
| Abducted | 4.118 (± 2.848)# | 2.647 (± 2.572)# | 0.650 (± 0.813)* | 0.800 (± 0.834)* |

LSS group: Lumbar spinal stenosis group; GMe: Gluteus medius; nQF: Normalized quadriceps femoris; TFL: Tensor fascia lata

*: P-Value <0.05 in comparison within each group

#: P-Value <0.05 in comparison to control group

Moreover, a wider stride width affected the FTA, shifting its degree to a more varus formation, increasing the loads on medial knee joints, which are known as a common location for osteo-arthritis in an elderly population. Although this tendency was not statistically verified in the LSS group, the control group showed significant changes toward a varus formation. The reason of insignificance in the LSS group might be related to the limited coronal motion of knee joints due to already developed OA. Lastly, when stride width deviated from its usual width, participants reported higher degrees of discomfort in both the hip and the medial knee areas.

Although the widening of stride width was not statistically significant, the changes in sEMG and FTA suggest a significant resemblance between the normal gait in the LSS group and abducted gaits in the control group. However, such a correlation seems to contradict with the hip abductor weakness exhibited by patients, which is a major symptom of LSS. This contradiction may be attributed to balance management. As a wider stride width is suggested to be

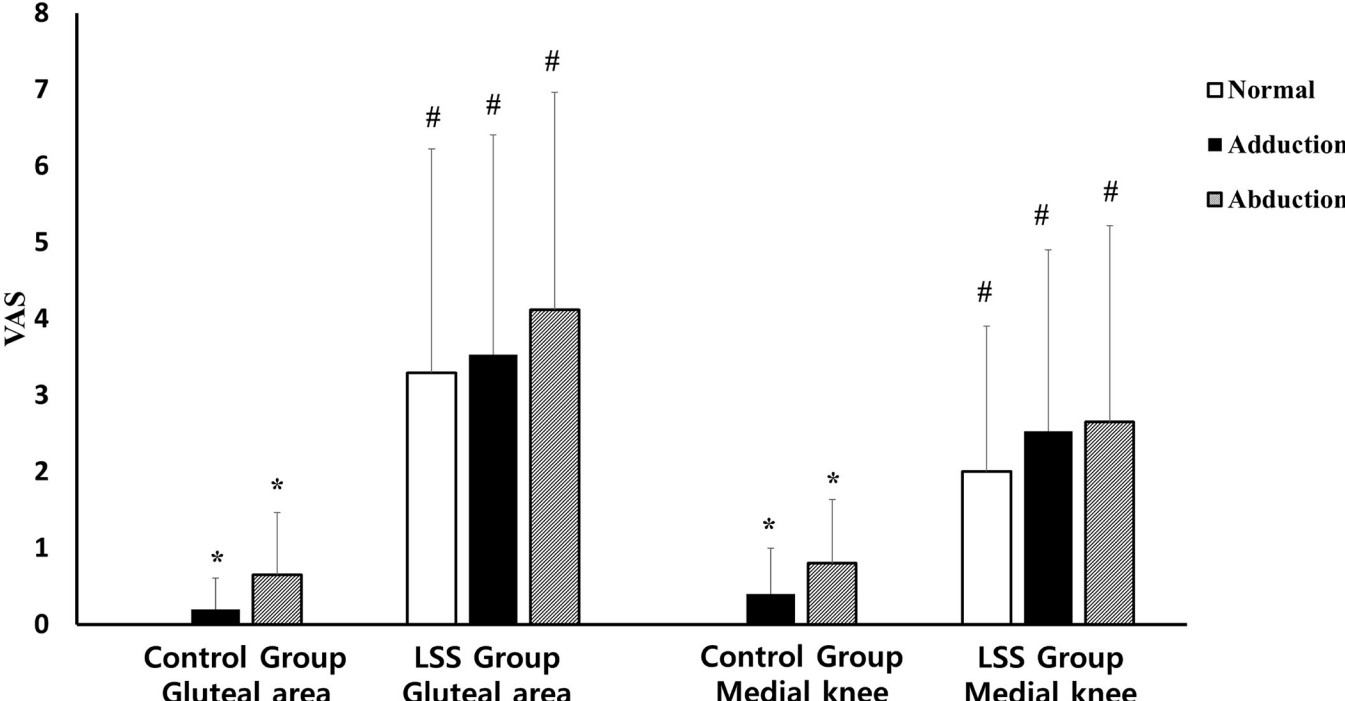

**Fig 4. This figure shows the discomfort level according to the gait patterns.** The VAS scores were significantly higher during adducted and abducted gait in both the hip and the medial side of the knee than the normal gait (p-value < 0.05).

related to enhanced balance[31], balance maintenance could be a reasonable explanation for the stride width observed in the LSS group. Moreover, a wider stride width is commonly observed in older populations[30, 32, 33].

Moreover, the sEMG activation pattern of QF supports this notion. When the control group was instructed to walk with wider strides, their QF sEMG signals decreased, while sEMG of GMe and TFL increased; however, when the LSS group was asked to conduct the abducted gait, their QF sEMG signals increased, together with the signals of GMe and TFL. This suggests that LSS patients may have limited function to control each muscle, require more effort to maintain balance, and thus, need to recruit more muscles. In other words, while healthy individuals require less QF activation during the abducted gait, LSS patients may need not only higher activation of the hip abductors, but also higher activation of QF (Table 3).

Also notable was the fact that the LSS group required higher activation of the hip abductor muscles for the same amount of abducted gait than the control group. Since stride width for abducted gait was designated for all participants, the extent of abduction was equivalent. Thus, LSS patients exhibited inefficient hip abductor use, and this was revealed in the form of over-firing of the hip abductor muscles. A possible interpretation would be that the over-firing of such muscles may be to compensate for an impaired balance.

Fig 5 shows the schematic diagram of a single limb support phase in the normal (A, d1) and abducted gait pattern (B, d1'). Widening of stride width in LSS patients despite abductor weakness implies that additional muscle recruitment (AB F', longer arrow) may be needed to maintain balance (B). Femorotibial angle (FTA) changed to the varus in an abducted gait pattern (B, FTA'). Such changes in the knee joint can be associated with knee osteoarthritis because when the knee joints are in a varus position, the loading of weight is focused on the medial side of the knee joint.

Furthermore, the alteration of gait patterns found in LSS patients and elderly population during the abducted gait shows the potential for stride width changing the knee alignment in these subjects.[29, 30] This is not only because LSS patients' knee joints tend to be closer to the varus during gait, but also because FTA changed when stride width was altered.[34] However, this study did not collect previous medical histories regarding knee osteoarthritis. Therefore, the causality between knee osteoarthritis and LSS cannot be drawn from this study. For clarification, long term follow-up studies should be conducted. Nevertheless, this study, to the best of our knowledge, is the first study to elucidate knee coronal angle changes in LSS patients, proposing a relationship between gait pattern and knee osteoarthritis from a biomechanical point of view.

## Study limitations

The control group was not age-matched; and due to the small sample size, the intragroup analysis based on gender was not possible. A follow-up study is required to evaluate the stride width according to age. However, this study evaluated three gait patterns such as normal, adducted and abducted gait patterns within each group. The differences in the three gait patterns in each group are thought to have contributed to the conclusion. This was a single session experiment; thus, long-term follow-up studies are needed to clarify the causality among LSS, joint angle, and knee osteoarthritis. Lastly, we have to consider the individual variation of sEMG.

## Conclusions

The patients with spinal stenosis showed a wider stride width compared to the control group. With a wider stride width, the relative activation of the hip abductors increased, and FTA

(A)　　　　　　　　　　　　　　　　　　　　　　(B)

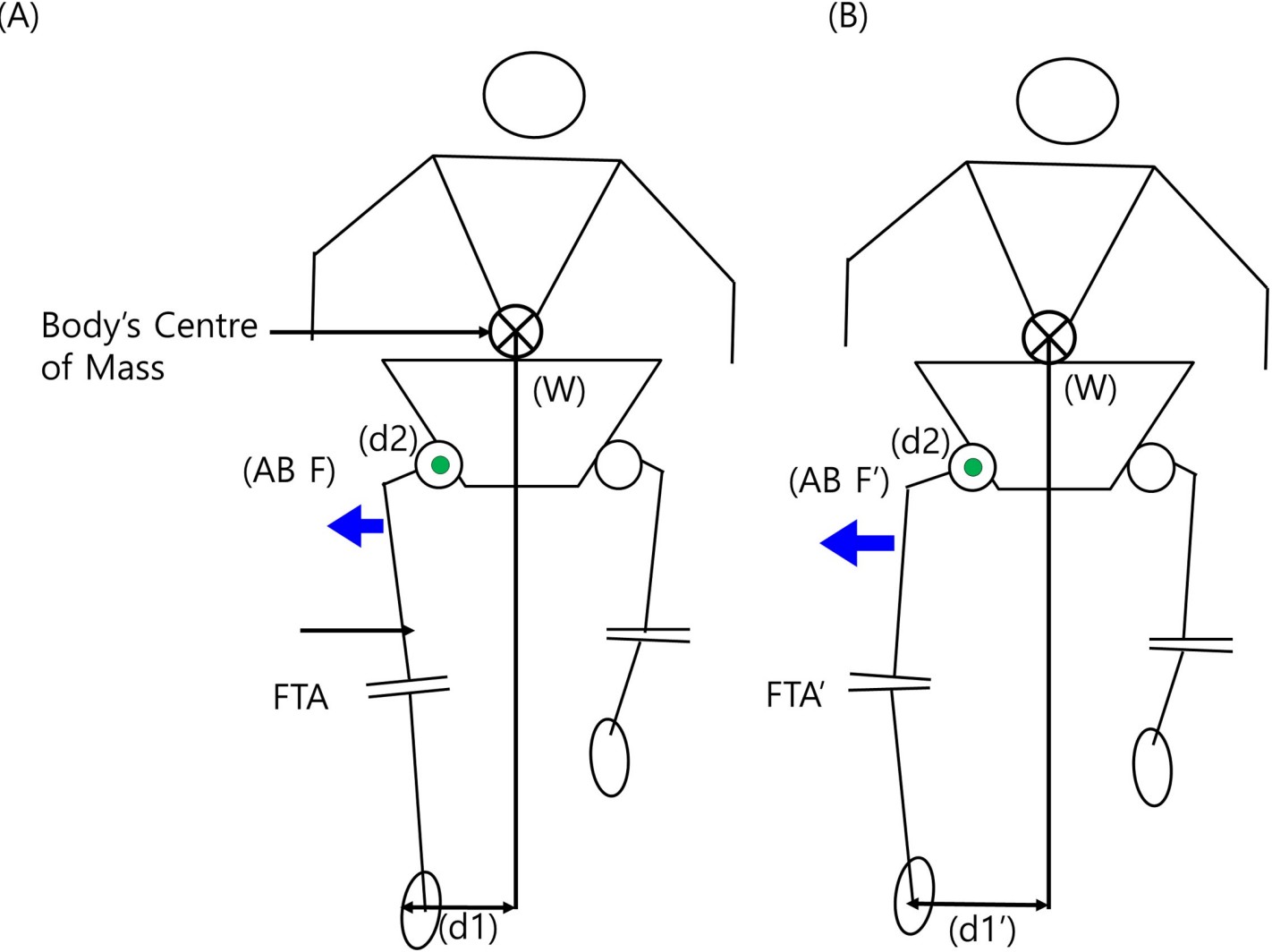

**Fig 5.** These figures show the schematic diagram of a single limb support phase in the normal (A, d1) and abducted gait pattern (B, d1'). Widening of stride width in LSS patients despite abductor weakness suggests that additional muscle recruitment (AB F', longer arrow) may be needed to maintain balance (B). Femorotibial angle (FTA) changed to the varus in an abducted gait pattern (B, FTA'). Furthermore, such a distinctive gait pattern exerts increased loading on the medial knee, relating to the escalated risk of degenerative knee osteoarthritis.

became closer to the varus. The VAS scores of the hip and medial side of the knee joint also increased with a widened stride width. The same tendency was observed in LSS patients compared with healthy individuals. Widening of stride width in LSS patients, despite abductor weakness, may indicate additional muscle recruitment for balance maintenance. Furthermore, such distinctive gait pattern exerts increased loading on the medial knee.

## Supporting information

**S1 Data.**
(XLSX)

## Acknowledgments

The authors thank the Medical Research Collaborating Center at Seoul National University Bundang Hospital for statistical analyses.

## Author Contributions

**Conceptualization:** Joonyoung Jang, Ju Seok Ryu.

**Data curation:** Jin Ju Kim, Han Cho.

**Formal analysis:** Jin Ju Kim, Han Cho.

**Investigation:** Jin Ju Kim, Han Cho, Yulhyun Park, Jung Woo Kim.

**Methodology:** Yulhyun Park, Joonyoung Jang.

**Supervision:** Joonyoung Jang, Ju Seok Ryu.

**Writing – original draft:** Jin Ju Kim, Han Cho.

**Writing – review & editing:** Ju Seok Ryu.

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
