## [Decision Letter · Decision Letter 0]

27 Nov 2019

PONE-D-19-20223

Biomechanical Influence of Gait Patterns on Knee Joint: Kinematic & EMG analysis

PLOS ONE

Dear Ju seok Ryu,

Thank you for submitting your manuscript to PLOS ONE. After careful consideration, we feel that it has merit but does not fully meet PLOS ONE’s publication criteria as it currently stands. Therefore, we invite you to submit a revised version of the manuscript that addresses the points raised during the review process.

We would appreciate receiving your revised manuscript by 03 Genuary. To enhance the reproducibility of your results, we recommend that if applicable you deposit your laboratory protocols in protocols.io, where a protocol can be assigned its own identifier (DOI) such that it can be cited independently in the future. For instructions see: http://journals.plos.org/plosone/s/submission-guidelines#loc-laboratory-protocols

We look forward to receiving your revised manuscript.

Kind regards,

Riccardo Di Giminiani

Academic Editor

PLOS ONE

Journal Requirements:

This research was supported by a grant of the Korea Health Technology R&D Project through the Manuscript Click here to download Korea Health Industry Development Institute (KHIDI), funded by the Ministry 23 of Health & Welfare, Republic of Korea (grant number : HI18C1169)

Please remove any funding-related text from the manuscript and let us know how you would like to update your Funding Statement. Currently, your Funding Statement reads as follows: Not applicable

Not applicable

Reviewers' comments:

Reviewer's Responses to Questions

**Comments to the Author**

1. Is the manuscript technically sound, and do the data support the conclusions?

Reviewer #1: Partly

Reviewer #2: Partly

2. Has the statistical analysis been performed appropriately and rigorously? 

Reviewer #1: No

Reviewer #2: I Don't Know

3. Have the authors made all data underlying the findings in their manuscript fully available?

Reviewer #1: Yes

Reviewer #2: Yes

4. Is the manuscript presented in an intelligible fashion and written in standard English?

Reviewer #1: Yes

Reviewer #2: No

5. Review Comments to the Author

Reviewer #1: Dear corresponding Author thanks for your submission. Please check these comments:

- The age diffrence between the two groups is significant. How can you be sure that this different is not the cause of different walking style? It is a sample problem.

- Please can you show information how you measured the stride width? You mean the horizontal distance between two feet? Or the stride length, in some part of the paper it is not clear.

- The statistical approach is very simple and I don't think that it is correct to use the paired T-Test for the comparison of different way to walk in the same group. This kind of procedure is called "repeated measure" and so the statistical approach should be different. Usually some correction factors are used (such as Bonferroni). Please check this point with a statistic specialist just to be sure.

- The results in table 2 are difficult to understand. Please can you explain how the sistem works to measure the FTA in varus and valgus. Moreover the table 2 in not clear in terms of significance: there are two different symbols and it is not clear the comparison. Please modify.

- The legend of figure 1A is wrong. The platform used is not a force plate but a pressure plate with a sensorized matrix.

- From the results it is not clear how did you use the stride width parameter. It is a very very interesting parameter but there is no explanation of the modification of this parameter in differents gait style. In the discussion paragraph you speak about stride width (line 259) but no data are presented in the results.

- Line 281: which part of the results allow you to write this sentence?

- So the practical application of this paper, as you wrote in conclusion, is to suggest to walk with abducted legs because the higher muscular recruitment? Can you explain better this point? I understand that the muscle recruitment is higher for sure but the pain and discomfort as well. So it seems a strange suggestion.

Reviewer #2: This paper requires major revision. It is not coherent or methodologically sound (at least as described).

Putting aside the missing definite article in the title, the title itself is misleading, especially since they state in their discussion that they cannot make conclusions about knee OA.

The topic is poorly explained and background on both clinical and basic research is basically non-existent.

The hypotheses are vague, asserting weakness and its effect on gait and on the knee and on knee OA. They are not specific. Moreover, the argument for why this affects the knee rather than the hip is not stated clearly or convincingly. A causative effect is simply asserted and not explained in a way that makes it predictable.

There is no real discussion in the introduction of the role played by abductors in normal gait and how the different gaits really worked. Is it not the case, that the patterns of EMG are very predictable across gaits. So this is really a question of comparisons between LSS and control. But even then the results are reasonably predictable.

I need clearer hypotheses and clearer statements about why this data is being pursued, aside from that it hasn't been collected before.

The adducted gait is intense and un-natural. I can only envision it as a shuffle. I could use more description of this gait to justify its inclusion.

The choice of gaits leads to obvious associations. For example, it makes sense a priori that stride width would require more abductors. It also makes sense that stride width changes knee alignment.

How do you know you got EMG from lesser gluteals. I believe you, but would like to know how you verified that was the case in detail.

I have no idea why the EMG was normalized to QF. Please expain the logic for this. It doesn't make sense to me at this point. I also worry that some of the patient group has OA already and that QF weakness and OA are correlated. So this may lead to weir values.

Much of the discussion addresses complexities and contradicitions in the results. But these challenges are unexpected to the reader because they are are poorly laid out in the introduction.

The discussion does a more thorough job setting up the problem and much of it could have appeared in the intro. But also in the discussion we learn that the data necessary to test at least one of the hypotheses has not been collected. "Therefore, the causality between knee osteoarthritis and LSS cannot be drawn from this study." This was a purpose of the study.

Figure 4 seems to be mislabeled.

6. PLOS authors have the option to publish the peer review history of their article (what does this mean?). If published, this will include your full peer review and any attached files.

Reviewer #1: No

Reviewer #2: No

---

## [Author Response · Author response to Decision Letter 0]

19 Dec 2019

Dear reviewers,

Hello,

I did my best to answer your questions.

I consulted to a statistician and correct the exact statistics.

The present study performed the gait analysis and EMG analysis in the coronal plane.

These methods are relatively new techniques and rarely performed previously.

Therefore, we expect this study to have a huge impact on medicine in the future.

I responded each point raised by editor and reviewers in "response to reviewer file".

I hope positive response.

Thank you.

---

## [Decision Letter · Decision Letter 1]

6 Feb 2020

PONE-D-19-20223R1

Biomechanical Influence of Gait Patterns on Knee Joint: Kinematic & EMG analysis

PLOS ONE

Dear Dr. Ryu,

Thank you for submitting your manuscript to PLOS ONE. After careful consideration, we feel that it has merit but does not fully meet PLOS ONE’s publication criteria as it currently stands. Therefore, we invite you to submit a revised version of the manuscript that addresses the points raised during the review process.

ACADEMIC EDITOR: The issues raised by reviewer should be addressed adequately before considering the manuscript for pubblication. 

We would appreciate receiving your revised manuscript by Mar 22 2020 11:59PM. To enhance the reproducibility of your results, we recommend that if applicable you deposit your laboratory protocols in protocols.io, where a protocol can be assigned its own identifier (DOI) such that it can be cited independently in the future. For instructions see: http://journals.plos.org/plosone/s/submission-guidelines#loc-laboratory-protocols

We look forward to receiving your revised manuscript.

Kind regards,

Riccardo Di Giminiani

Academic Editor

PLOS ONE

Reviewers' comments:

Reviewer's Responses to Questions

**Comments to the Author**

1. If the authors have adequately addressed your comments raised in a previous round of review and you feel that this manuscript is now acceptable for publication, you may indicate that here to bypass the “Comments to the Author” section, enter your conflict of interest statement in the “Confidential to Editor” section, and submit your "Accept" recommendation.

Reviewer #1: (No Response)

2. Is the manuscript technically sound, and do the data support the conclusions?

Reviewer #1: (No Response)

3. Has the statistical analysis been performed appropriately and rigorously? 

Reviewer #1: (No Response)

4. Have the authors made all data underlying the findings in their manuscript fully available?

Reviewer #1: (No Response)

5. Is the manuscript presented in an intelligible fashion and written in standard English?

Reviewer #1: (No Response)

6. Review Comments to the Author

Reviewer #1: 1) the age problem in the sample is still present

2) I know the sotfware you use and it doesn't measure the width, moreover it is not showed in the figure 1A as you write.

7. PLOS authors have the option to publish the peer review history of their article (what does this mean?). If published, this will include your full peer review and any attached files.

Reviewer #1: No

---

## [Author Response · Author response to Decision Letter 1]

19 Feb 2020

Dear reviewers,

Thank you for your comments.

In 2nd revision, we answered all of the questions in response to reviewer_v2 file.

Most important question was the method to measure stride width. 

In the present study, the raw data of the contact points made during each gait cycle was exported as Microsoft Excel file(.XLS format) according to manual specified by the FreeStep program. The width of each cell was 0.5cm. By counting the medial-lateral distance between the heels in terms of cell number, we could obtain the stride width. 

Thank you for your comments.

---

## [Decision Letter · Decision Letter 2]

11 May 2020

Biomechanical Influence of Gait Patterns on Knee Joint: Kinematic & EMG analysis

PONE-D-19-20223R2

Dear Dr. Ryu,

We are pleased to inform you that your manuscript has been judged scientifically suitable for publication and will be formally accepted for publication once it complies with all outstanding technical requirements.

With kind regards,

Riccardo Di Giminiani

Academic Editor

PLOS ONE

Additional Editor Comments (optional):

Reviewers' comments:

Reviewer's Responses to Questions

**Comments to the Author**

1. If the authors have adequately addressed your comments raised in a previous round of review and you feel that this manuscript is now acceptable for publication, you may indicate that here to bypass the “Comments to the Author” section, enter your conflict of interest statement in the “Confidential to Editor” section, and submit your "Accept" recommendation.

Reviewer #1: All comments have been addressed

2. Is the manuscript technically sound, and do the data support the conclusions?

Reviewer #1: (No Response)

3. Has the statistical analysis been performed appropriately and rigorously? 

Reviewer #1: (No Response)

4. Have the authors made all data underlying the findings in their manuscript fully available?

Reviewer #1: (No Response)

5. Is the manuscript presented in an intelligible fashion and written in standard English?

Reviewer #1: (No Response)

6. Review Comments to the Author

Reviewer #1: (No Response)

7. PLOS authors have the option to publish the peer review history of their article (what does this mean?). If published, this will include your full peer review and any attached files.

Reviewer #1: No

---

## [Editor Report · Acceptance letter]

18 May 2020

PONE-D-19-20223R2 

Biomechanical Influences of Gait Patterns on Knee Joint: Kinematic & EMG Analysis 

Dear Dr. Ryu:

I am pleased to inform you that your manuscript has been deemed suitable for publication in PLOS ONE. Congratulations! Your manuscript is now with our production department. 

With kind regards,

on behalf of

Prof. Riccardo Di Giminiani 

Academic Editor

PLOS ONE